# Evaluating the Necessity and Impact of Cardiac Imaging on Breast Cancer Care in Northwestern Ontario

**DOI:** 10.3390/cancers17121909

**Published:** 2025-06-08

**Authors:** Hannah Shortreed, Megan Clark, Husam Abdel-Qadir, Rabail Siddiqui, Olexiy Aseyev

**Affiliations:** 1Department of Undergraduate Medical Education, Northern Ontario School of Medicine University, Thunder Bay, ON P7B 7A5, Canada; megclark@nosm.ca (M.C.); olexiy.aseyev@tbh.net (O.A.); 2Division of Cardiology and Department of Medicine, Women’s College Hospital and University Health Network, Toronto, ON M5S 1B2, Canada; husam.abdel-qadir@wchospital.ca; 3Department of Medicine and Institute of Health Policy, University of Toronto, Toronto, ON M5S 1A1, Canada; 4Department of Clinical Research Services, Thunder Bay Regional Health Research Institute, Thunder Bay, ON P7B 7A4, Canada; rabail.siddiqui@tbh.net; 5Division of Medical Oncology and Department of Medicine, Thunder Bay Regional Health Sciences Centre, Thunder Bay, ON P7B 7A5, Canada; 6Division of Medical Oncology and Department of Medicine, Cancer Care Northwest, Thunder Bay, ON P7B 7A5, Canada

**Keywords:** breast cancer, cardiac imaging, cardiotoxicity, anthracyclines, trastuzumab, resource allocation

## Abstract

Women undergoing treatment for breast cancer often receive cardiac imaging before starting therapy to assess their heart health, as some treatments can increase the risk of heart problems. However, the guidelines for when and how often to perform these tests are not well-defined. This study examines the effectiveness of cardiac imaging under different treatment regimens and identifies factors that may predict when imaging is most useful. By analyzing past cases, the authors found that some treatment groups had a higher likelihood of imaging results leading to changes in care, while others had lower likelihood despite more frequent imaging. Understanding these patterns can help doctors make informed decisions about ordering cardiac imaging. This may improve patient care and resource use, especially in regions with limited healthcare access.

## 1. Introduction

Over 25,000 women are diagnosed with breast cancer annually in Canada, with approximately 10,000 of these cases occurring in Ontario alone [1]. Despite improvements in screening and treatment [2,3,4,5], breast cancer is the second leading cause of cancer-related mortality in Canadian women, following lung cancer [1]. A key contributor to morbidity and mortality is treatment-induced cardiotoxicity and cardiovascular disease (CVD) [6,7]. In Ontario, patients receiving chemotherapy for early-stage breast cancer were found to have a threefold increased risk for serious cardiovascular events (i.e., heart failure [HF]) compared to age-matched controls [8]. As a result, CVD is a serious complication in older breast cancer patients, responsible for 16.3% of deaths and surpassing breast cancer as the leading cause of death at a 10-year follow-up for those with previous CVD [9,10].

Cardiac imaging modalities, including echocardiography, multiple gated acquisition scanning (MUGA), and cardiac magnetic resonance imaging (CMR), can identify early signs of HF, particularly reduced left ventricular ejection fraction (LVEF) [11,12,13]. For patients receiving anthracyclines, trastuzumab, or a combination of both, baseline cardiac imaging and monitoring of LVEF throughout and after therapy is essential for identifying those at risk of developing treatment-related HF [12,13]. Baseline cardiac imaging for breast cancer patients is specifically recommended in the Clinical Practice Guidelines listed by the National Comprehensive Cancer Network (NCCN) [14]. Once identified, reduced LVEF can be managed to prevent HF progression. These include introducing or increasing the dose of cardioprotective medications (i.e., statins or beta-blockers), consulting new care providers (i.e., cardiology and cardio-oncology specialists), and adjusting chemotherapy regimens [15,16,17].

Despite the recognized importance of LVEF monitoring, the frequency and criteria for imaging are often based on expert consensus rather than robust clinical data [14,18]. Consequently, imaging practices vary by treatment regimen [19]. For example, patients receiving trastuzumab typically undergo imaging every three months per FDA labelling, regardless of HF risk factors [14,20], while those on anthracyclines may receive less frequent imaging, even if they are more likely to develop HF [19]. Overall, these risk factors are associated with a significantly increased risk of major cardiac events, regardless of treatment type [19]. This indicates that there is discordance between the current utilization of cardiac imaging and a patient’s risk of developing HF or CVD while receiving potentially cardiotoxic therapy [19].

This study aimed to analyze local data to determine the impact of cardiac imaging on treatment outcomes for breast cancer patients receiving anthracyclines, trastuzumab, or combination therapy and offer evidence-based guidance for ordering physicians. In doing so, we hope to optimize cardiac imaging to enhance resource allocation in Northwestern Ontario (NWO), a historically resource-limited region, and improve patient outcomes [21,22,23].

## 2. Materials and Methods

### 2.1. Study Population

This retrospective cohort study included all female patients seen at a regional cancer centre in NWO who were treated with anthracyclines and/or trastuzumab for newly diagnosed breast cancer between 1 June 2014 and 31 December 2017. Exclusion criteria were as follows: neither anthracycline and/or trastuzumab as part of systemic therapy, missing complete records regarding anthracycline and/or trastuzumab therapy, absence of imaging tests (echocardiography, MUGA, or CMR) ordered following diagnosis until one-year post-treatment, missing complete records of imaging tests ordered (Figure 1).

Demographic and breast-cancer-related data were collected from electronic medical records and included the following: age, menopausal status (pre-, peri-, and postmenopausal), body mass index (BMI), cancer type (invasive ductal carcinoma, lobular carcinoma, or ductal carcinoma in situ), cancer location (right/left), tumour stage (I–III), receptor status (estrogen, progesterone, and human epidermal growth factor receptor 2 [HER2]), and breast cancer gene (BRCA) status.

Medical history prior to beginning treatment was recorded and included pre-treatment/earliest Eastern Cooperative Oncology Group (ECOG) score, pre-treatment/earliest Karnofsky score, pregnancy status, and smoking status. Other data such as diagnosis of diabetes, hypertension, high cholesterol, coronary artery disease (CAD), HF, valvular disease, atrial fibrillation, chronic obstructive pulmonary disease (COPD), peripheral vascular disease (PVD), stroke/transient ischemic attack (TIA), chronic kidney disease (CKD), previous breast cancer, previous other cancer, and previous treatment with anthracyclines or radiation was also collected.

Information regarding the cardiac imaging tests performed included the method of cardiac imaging test (echocardiography, MUGA, or CMR), time-point (baseline or after initiating systemic therapy), the referring physician (oncologist or non-oncologist), ordering mechanism (routine or prompted by clinical factors), and LVEF (%).

This study received Research Ethics Board approval from the Thunder Bay Regional Health Sciences Centre (#100175).

### 2.2. Outcomes

The primary outcome was to determine whether there was a significant variation in the yield of cardiac imaging based on different treatment regimens involving anthracycline and/or trastuzumab. Yield was defined as the total number of changes arising from the total number of scans. By this definition, one scan can produce multiple changes to reflect the quantitative yield of each scan. Events classified as a change in clinical care include chemotherapy changes (change in agent or temporary/permanent discontinuation), referral to another care provider (outpatient referral to cardiologist/non-cardiologist or hospital admission), and medication changes (new/increased doses of angiotensin-converting enzyme inhibitors, angiotensin II receptor blockers, beta-blockers, statins, spironolactone, or eplerenone). Secondary outcomes include examining the correlation between patient-specific variables or features of cardiac imaging and changes in clinical care. This will determine if there are any predictors of a higher yield of cardiac imaging. Variables examined include the patient’s medical history before treatment (i.e., smoking status), demographic features (i.e., age and BMI), and the features of cardiac imaging (i.e., routine vs. prompted).

### 2.3. Statistical Analysis

Patients were grouped into three cohorts based on their treatment regimen: trastuzumab-only (cohort A), anthracycline-only (cohort B), and combined therapy (cohort C). Each cohort underwent individual statistical analysis. The primary outcome (the yield of cardiac imaging) was expressed as the proportion (%) of clinically actionable changes in care (either proactive or reactive) arising from all conducted imaging tests. Due to an insufficient amount of data for adequate statistical power, cohort A was excluded from quantitative primary analysis. The significance of variation amongst individual cohorts (i.e., cohort B vs. cohort C) was determined using a Z-test for proportions with the Bonferroni correction to adjust for multiple comparisons. For those cohorts with proportions of changes/scans greater than 10%, the primary outcome was further examined by categorizing the types of changes (i.e., change in chemotherapy, care provider, or medication). This was performed by calculating the proportion of each category of change relative to the total number of changes (i.e., if 3 out of 10 changes were due to X, X would account for 30% of total changes). Analysis of the significance of the variation between these categories was determined using a permutation test due to small sample sizes. As a continuation of the primary analysis, the number of scans per patient was determined for each cohort, and the significance of these ratios was determined using the Kruskal–Wallis test (overall variation amongst cohorts) and Dunn’s test (for variation amongst individual cohorts).

For analysis of the secondary outcomes, each variable that depicted categorical data (i.e., smoking status) was organized into a contingency table that included each category of the variable (i.e., current smoker, former smoker, non-smoker) and its association with either a “change” or “no change” in clinical care. In the case of incomplete data, missing values were retained and categorized as “unknown” (i.e., “unknown menopausal status”) in the analysis to preserve sample size and account for the presence of missing data without introducing bias through estimation. This process was repeated for each type of change (i.e., chemotherapy, care provider, and medication) and combined into a cumulative contingency table that underwent subsequent analysis using a Fisher’s exact test to assess whether there was a significant correlation between the variable and a change in clinical care. The significance of secondary outcomes for each variable was assessed using a Fisher’s exact test due to the low expected values within the tables (>20% of expected cell counts were <5). For secondary analysis of patient variables with continuous data (age at time of diagnosis, BMI, and LVEF), normality testing with a Shapiro–Wilk test was performed owing to small and unequal sample sizes. If the data was normally distributed, then parametric testing was pursued via an independent two-sample T-test to determine if there was a significant correlation between the variables and a change in clinical care. If the data was not normally distributed, then non-parametric testing was pursued via a Mann–Whitney U test. Due to an insufficient amount of data for adequate statistical power, cohort A was excluded from the secondary analysis.

For all analyses, *p* < 0.05 was considered statistically significant. The confidence interval (CI) of proportional data (i.e., what proportion of current smokers experienced a change in clinical care) was calculated with 95% CI’s using the Wilson score interval due to small sample sizes [24]. The 95% CI’s for all mean values (i.e., scans per patient) were calculated using the t-score distribution due to small sample sizes [25]. Calculations and analyses of statistical significance were performed using Google Colaboratory [25].

## 3. Results

### 3.1. Patient and Cardiac Imaging Characteristics

A total of 93 patients met the exclusion criteria for this analysis. Of these, 3 underwent treatment with trastuzumab (cohort A), 60 with anthracyclines (cohort B), and 30 received combined therapy (cohort C). Treatment modalities included surgery (lumpectomy and mastectomy), anthracycline, trastuzumab, and endocrine therapy (oophorectomy, gonadotropin-releasing hormone (GnRH) agonists, tamoxifen, and aromatase inhibitors). The baseline demographic characteristics and disease characteristics of each cohort are summarized in Table 1. See Appendix A for patients’ medical history prior to starting treatment and Appendix A for information regarding the characteristics of cardiac imaging.

### 3.2. The Yield of Cardiac Imaging Based on Treatment Regimen

With respect to the primary outcome, there was a significant variation (*p* = 0.013) between cohort B and C for the yield of cardiac imaging, with cohort B having the largest yield (13.33%; 95% CI, 7.41–22.83%) (Table 2). Of these, 30.00% were due to changes in chemotherapy (*n* = 3; 95% CI, 10.78–60.32%), 40.00% were due to changes in the care provider (*n* = 4; 95% CI, 16.82–68.73%), and 30.00% were due to medication changes (*n* = 3; 95% CI, 10.78–60.32%). There was no significant variation in the types of change at *p* < 0.05 (*p* = 1.00). By comparison, cohort C had a yield of 4.17% (95% CI, 1.92–8.79%) and had the most scans per patient (4.80 per patient; 95% CI, 4.18–5.42), followed by cohort B (1.25 per patient; 95% CI, 1.10–1.40). Due to the very small sample size in cohort A, findings from this group are presented as a pilot exploration and are intended for qualitative insight only, not statistical comparison. Qualitatively, cohort A had the second highest yield (7.14%; 95% CI, 1.27–31.47%) and had the second most scans per patient (4.67 per patient; 95% CI, −0.50–9.84). These results are summarized in Table 2. Overall, the number of scans per patient was significantly different amongst cohorts (*p* < 0.001) (Table 2).

### 3.3. Predictive Variables for a Higher Yield of Cardiac Imaging

Secondary outcomes included examining the correlation between patient-specific variables or features of cardiac imaging and changes in clinical care to determine if there were any predictors of a higher yield of cardiac imaging. Variables examined included the patient’s medical history before treatment (i.e., smoking status), demographic features (i.e., age and BMI), and the ordering mechanism of cardiac imaging (i.e., routine vs. prompted). Cohort A was excluded from the secondary analysis owing to an insufficient number of patients (*n* = 3), preventing the interpretation of any significant conclusions at *p* < 0.05.

In cohort B, several predictive variables were found to be of statistical significance, including ECOG score, menopausal status, diabetic status, CAD status, PVD status, as well as the ordering mechanism, study time-point, referring physician, and the LVEF associated with the cardiac imaging resulting in a change in care (Table 3, Figure 2). Specifically, a higher ECOG score was significantly associated with a greater likelihood of a change in clinical care (*p* < 0.001), with the strongest associations observed for patients with ECOG 2 (OR = 35.64, and 64.27 compared to ECOG 1, and ECOG 0, respectively; the variation in yield was not significant compared to ECOG unknown, OR = 4.57). Similarly, patients with unknown menopausal status were significantly more likely to experience a change in care compared to other menopausal statuses (*p* < 0.001; OR = 28.37, 22.58, and 16.53 compared to pre-, peri-, and post-menopausal patients, respectively). Additionally, being diabetic (*p* = 0.001; OR = 12.11), having CAD (*p* = 0.048; OR = 5.20), and having PVD (*p* < 0.001; OR = ∞) before beginning treatment were associated with a greater likelihood of a change in clinical care following cardiac imaging (Figure 2). Among patients who had a change in care following imaging, all were found to have at least two predictive variables; by comparison, the yield of imaging in patients with no predictive variables was 0% (*n* = 5, 95% CI, 0.00–43.45%).

Regarding imaging, those patients who underwent cardiac imaging prompted by clinical factors (*p* < 0.001; OR = 31.71), had imaging completed during chemotherapy (*p* < 0.001; OR = 200.22 compared to baseline imaging, respectively; the variation in the yield of cardiac imaging did not differ significantly when compared to imaging post-chemotherapy, OR = 2.23), and were referred for imaging by a critical care physician (*p* < 0.001; OR = 251.22 compared to oncologist referrals; the variation in yield was not significant when compared to family physicians or cardiologists, OR = 21.00 and 49.00, respectively) were more likely to have a change in care (Figure 2). Moreover, average LVEF was significantly lower for those patients who experienced a change in care following cardiac imaging compared to those who did not (*p* = 0.02, T-statistic = −3.53; 43.67% ± 13.32 vs. 64.96 ± 6.96, respectively) (Table 3). Finally, there was no significant variation in the yield of imaging reported for patients who underwent echocardiography compared to those who had a MUGA scan (*p* = 0.67). See Appendix A for the complete list of pre-treatment and imaging variables examined for this cohort.

In cohort C, the predictive variables associated with a change in clinical care following cardiac imaging included: smoking status, atrial fibrillation, COPD, and LVEF (Figure 2). Specifically, patients who were current smokers (*p* = 0.005; OR = 14.00 and 18.00 compared to former and non-smokers, respectively), those who had atrial fibrillation (*p* = 0.008; OR = ∞), and those with COPD (*p* = 0.002; OR = 53.00) had a greater likelihood of experiencing a change in clinical care (Figure 2). Moreover, the average LVEF was significantly lower for those patients which experienced a change in clinical care following cardiac imaging compared to those who did not (*p* < 0.001, T-statistic = −6.90; 44.75% ± 4.79 vs. 62.95 ± 7.72, respectively) (Table 3). Amongst patients who had a change in care following imaging, all were found to have at least two predictive variables; by comparison, the yield of imaging in patients with no predictive variables was 0% (*n* = 3, 95% CI, 0.00–56.15%). Finally, there was no significant variation in the yield of imaging reported for patients who underwent echocardiography compared to those who had a MUGA scan (*p* = 1.0). See Appendix A for the complete list of pre-treatment and imaging variables examined for this cohort.

## 4. Discussion

This study’s findings highlight the variability in the clinical utility of cardiac imaging in breast cancer patients treated with anthracyclines and/or trastuzumab. Cardiac imaging, while essential for monitoring cardiotoxicity, demonstrated significantly different yields across cohorts (*p* = 0.013). Cohort B had the highest imaging yield at 13.33% (95% CI 7.41–22.83%), compared to 4.17% (95% CI 1.92–8.79%) for patients in cohort C (Table 2). Moreover, patients in cohort B received fewer imaging studies per patient (1.25, 95% CI 1.10–1.40) compared to cohort C (4.67, 95% CI −0.50–9.84) (*p* < 0.001), contributing to this cohort’s higher yield (Table 2). This discrepancy between imaging yield and resource utilization challenges the efficiency of current imaging protocols. Further studies are warranted to stratify yield more granularly to distinguish between types of change (i.e., proactive versus reactive) as well as examine the clinical impact of change in care on patient outcomes.

Importantly, patients in cohort A were excluded from quantitative primary analysis due to their small sample size and low statistical power [26]. The results of its analysis for qualitative purposes reveal that patients in cohort A had the second highest yield (7.14%; 95% CI, 1.27–31.47) and had the second most scans per patient (4.67 per patient; 95% CI, −0.50–9.84) when compared to patients in other cohorts.

In addition to the primary outcome, predictors of higher imaging yield differed depending on the treatment regimen. Among patients in cohort B, significant variables included a higher ECOG score (*p* < 0.001), unknown menopausal status (*p* < 0.001), diabetes (*p* = 0.001), CAD (*p* = 0.048), PVD (*p* < 0.001), cardiac imaging prompted by clinical factors (*p* < 0.001), imaging performed during chemotherapy (*p* < 0.001), and referrals from critical care physicians (*p* < 0.001) (Figure 2). In contrast, patients in cohort C demonstrated a higher yield for cardiac imaging if they were current smokers (*p* = 0.005), had atrial fibrillation (*p* = 0.008), or had COPD (*p* = 0.002) (Figure 2). The only shared variable for higher cardiac imaging yield between the two groups was lower LVEF (*p* = 0.02 and *p* < 0.001, respectively) (Table 3). This finding aligns with the established role of LVEF as a primary indicator of cardiac function and a critical factor in managing patients at risk for HF [12,13,27,28]. In the context of this study, low LVEF likely represents the main determinant in providers’ decision to intervene in these patients’ care, accounting for why it is the sole shared variable between cohorts.

As with primary analysis, patients in cohort A were excluded from secondary analysis due to a very small sample size (*n* = 3), resulting in low statistical power [26]. Small and unequal cohort sizes (*n* = 60 for cohort B and *n* = 30 in cohort C) represent a key limitation of this study, potentially affecting the validity of inter-cohort comparisons by reflecting size disparities rather than true variation [29,30]. Data collection is ongoing with larger sample sizes encompassing multiple hospitals in Ontario, and future analysis of this data will help address this limitation. Additional limitations include unaccounted confounders, such as cumulative anthracycline dose and prior chemotherapy exposure [31]. Furthermore, as this is a retrospective cohort study, there are inherent limitations in establishing causality versus correlation when describing significant results [32].

In contrast, strengths of this study include the emphasis on a clinically relevant population, analysis of multiple treatment regimens, and evaluation of diverse predictive variables to improve cardiac imaging yield. As the first of its kind in NWO, this study may enhance clinical practice, optimize the use of cardiac imaging in a resource-limited region, and improve patient outcomes [21,22,23].

To elaborate on our primary findings, our results indicate that patients in cohort B demonstrated the highest yield of cardiac imaging, despite having fewer scans per patient compared to those in cohort C. The reduced number of scans and, subsequently, higher yield, is likely due to the specific recommendations for cardiac imaging in patients undergoing treatment with different agents. Specifically, patients receiving trastuzumab are required to undergo LVEF measurement at baseline and every three months as recommended by the FDA and endorsed by clinical practice guidelines, such as those from the NCCN and the American Society of Clinical Oncology [14,20,33]. This results in a minimum of four scans per year in patients receiving trastuzumab [14,20,33]. In contrast, patients receiving anthracyclines undergo less frequent imaging based on current consensus documents; the 2024 NCCN guidelines recommend cardiac imaging within one year of the final anthracycline dose for patients with a significant cumulative anthracycline dose (≥250 mg/m^2^ of doxorubicin or equivalent) or a lower cumulative dose with one or more risk factors for HF [18]. This recommendation aligns with the results of one study, wherein 98% of cardiotoxicity cases occurred within one year of completion of treatment [34]. However, the indication for imaging in patients who are more likely to develop HF relies primarily on expert opinion, as the key studies cited in this rationale examined populations without significant cardiac risk factors or excluded patients with CAD risk factors through additional evaluations for ischemic aetiologies [34,35].

These expert recommendations are relatively recent and predate the study range of our cohort (2014–2017). Importantly, they also focus on routine imaging. According to our data, routine imaging accounted for 96.67% of scans in patients in cohort C, compared to 86.67% in cohort B (Appendix A). Additionally, we observed a significant correlation between imaging prompted by clinical factors and a higher yield of cardiac imaging in cohort B (*p* < 0.001). This may suggest that imaging for anthracycline-only patients was more often driven by clinical signs of HF or cardiac dysfunction, resulting in a significantly higher yield compared to routine imaging in patients receiving combined therapy.

This finding is consistent with prior expectations, as patients receiving anthracyclines typically undergo less routine imaging [18]. Consequently, significant imaging findings are more likely to be identified during clinically indicated tests. While the implications are not definitive, these findings support refining imaging guidelines for patients on anthracycline therapy. Future research is needed to evaluate whether a more proactive imaging approach could facilitate early detection of cardiotoxic effects before clinical symptoms arise, enabling timely intervention and potentially improving patient outcomes. Furthermore, by shifting from a reactive to a proactive imaging strategy, this approach may reduce long-term healthcare burdens, particularly in rural settings, by lowering costs associated with advanced cardiovascular disease and alleviating clinician burden.

With respect to patient-specific factors associated with a higher yield of cardiac imaging, our study identified minimal overlap between treatment cohorts (Figure 2), apart from decreased LVEF. The implications of this are unclear but could be attributable to variations in treatment regimens and the mechanisms of treatment-induced cardiotoxicity. Both anthracyclines and trastuzumab are independently recognized as risk factors for cardiotoxicity [6,8]. When used sequentially in the same patient, it follows that the cumulative risk of cardiotoxicity increases compared to using either agent alone. This was demonstrated in a population-based retrospective cohort study examining cancer therapy-related cardiac dysfunction in women with breast cancer, where sequential therapy with anthracyclines and trastuzumab was associated with a higher cumulative incidence of major cardiac events compared to either trastuzumab or anthracyclines alone [8]. Consequently, patients in our analysis who underwent combined therapy were likely at elevated risk of developing cardiotoxicity, potentially resulting in changes in their care following cardiac imaging. This elevated risk may have been independent of patient-specific factors. Using diabetes as an example, patients treated with both agents may have experienced cardiotoxicity due to the cumulative effects of the drugs, regardless of whether they had diabetes. Alternatively, diabetes may have contributed only minimally to the overall risk. While plausible, this hypothesis is only speculative in the absence of more robust analyses. Future research, including data from a trastuzumab-only cohort, is required to confirm these patterns.

Regarding individual cohorts, patient-specific factors significantly correlated with a higher yield of imaging in patients in cohort B included a higher ECOG score (ECOG = 2), unknown menopausal status, diabetes, CAD, and PVD (Figure 2). Additionally, imaging completed during chemotherapy, prompted by clinical factors, or referred by a critical care physician, was also associated with a significantly higher yield (Figure 2). The ECOG score is a tool used to assess a patient’s capacity to tolerate treatment based on their symptomology and activity level [36]. It measures their functional status, with higher scores indicating greater dysfunction and reduced ability to tolerate therapy [36]. Since cardiotoxicity can indicate impaired functional status, it is logical that a higher ECOG score correlates with a higher yield in imaging tests designed to diagnose and manage cardiac dysfunction [6,7]. While the ECOG score is a meaningful metric, unknown menopausal status is less informative in predicting which patients are likely to benefit from cardiac imaging. This finding likely reflects incomplete or missing data from chart reviews. Evidence shows that women’s risk of cardiovascular disease increases significantly after menopause, likely due to the loss of estrogen’s cardioprotective effects [37,38]. Consequently, postmenopausal women undergoing anthracycline therapy may have a higher yield of cardiac imaging due to their elevated disease risk. Future studies should investigate this hypothesis to provide better guidance to ordering physicians.

With respect to the other patient variables that were found to be associated with a higher yield in cohort B, conditions such as diabetes, CAD, and PVD are known to increase cardiovascular disease risk [39,40]. We hypothesize that these cardiovascular risk factors compounded the risk for treatment-induced cardiotoxicity in breast cancer patients, leading to a higher imaging yield. In contrast, other known risk factors, such as smoking and hypertension [41], were not significantly associated with a higher imaging yield. This may result from the small cohort size (*n* = 60), which could have resulted in a type II error or false negative [30].

Regarding imaging-specific variables, imaging prompted by clinical factors was previously discussed as being linked to a higher yield. However, imaging completed during chemotherapy and imaging ordered by a critical care physician were also associated with a higher yield in patients in cohort B (Figure 2). Imaging completed during chemotherapy appears to have had the greatest impact on care when compared with baseline imaging. Baseline imaging is less likely to be associated with a higher yield because patients with marked LVEF dysfunction (i.e., LVEF < 50%) at baseline are typically recommended alternative non-cardiotoxic treatments or cardioprotective therapy, as per the European Society of Medical Oncology consensus recommendations [42]. Conversely, studies suggest that the highest incidence of cardiotoxicity occurs within the first year after completing anthracycline therapy (up to 98% of cases), a finding that contrasts with our results [34]. This discrepancy may stem from our study design, as the yield of cardiac imaging was defined not as the incidence of treatment-induced cardiac toxicity but as the cumulative change in clinical care resulting from imaging findings. Patients experiencing a change in care may have exhibited early LVEF dysfunction that had not yet progressed to cardiotoxicity, leading to preventative treatment to limit further progression. Finally, the correlation between imaging ordered by a critical care physician and a higher yield of cardiac imaging likely reflects the patient population managed by these providers. Critical care physicians often care for patients experiencing the acute effects of cardiotoxicity, resulting in a higher pre-test probability for cardiotoxicity compared to imaging ordered by physicians working in less acute settings (i.e., oncologists).

By comparison, patient-specific factors significantly correlated with a higher yield of imaging in cohort C included current smoking, atrial fibrillation, and COPD (Figure 2). Smoking is a known cardiovascular risk factor, as is atrial fibrillation [41,43]. Additionally, evidence suggests that there is a correlation between COPD and cardiovascular disease, partially due to shared risk factors such as smoking and ageing, as well as shared pathophysiologic processes, including inflammation and arterial stiffness [44,45,46]. As with cohort B, it is hypothesized that these factors increased the risk of cardiotoxicity, resulting in a higher yield of cardiac imaging for breast cancer patients receiving combined therapy.

Concerning the broader applicability of our study, which is focused on a smaller, regional cancer centre in NWO, firm conclusions beyond this context are limited as this lies beyond the scope of our dataset. However, we acknowledge that related research in larger, urban centres provides useful points of comparison. For example, a retrospective study conducted at Princess Margaret Cancer Centre (Toronto, Canada) examined the utility of routine cardiac imaging in 448 breast cancer patients receiving adjuvant trastuzumab [47]. The study found that only 2.8% of routine imaging tests led to a change in patient care, compared to 27.8% of clinically prompted tests, suggesting limited clinical yield of routine imaging in this setting. Importantly, their findings were most applicable to patients at higher cardiac risk or earlier in treatment, and the study excluded lower-risk patients with insufficient prior imaging changes. In contrast, our study excluded trastuzumab-treated patients due to insufficient sample size, limiting our ability to directly compare findings. Nonetheless, both studies underscore the need for tailored imaging strategies and highlight the variability in imaging practices across care settings.

## 5. Conclusions

In conclusion, this study highlights the variability in the yield of cardiac imaging across different treatment cohorts, emphasizing its higher utility in patients receiving anthracycline-only therapy. The findings support reassessing current imaging protocols, particularly for patients receiving trastuzumab, where the frequency of routine scans does not correlate with a higher yield. Ordering physicians should consider patient-specific factors associated with a higher yield of cardiac imaging, which varies by treatment regimen. For patients receiving anthracycline-only therapy, factors linked to a higher imaging yield included a higher ECOG score, diabetes, CAD, PVD, lower LVEF, and imaging completed during chemotherapy, prompted by clinical factors, or ordered by a critical care physician. In contrast, for patients receiving combined therapy, current smoking, atrial fibrillation, COPD, and lower LVEF were associated with a higher yield. These variables can guide future imaging guidelines that support a proactive approach, particularly for patients receiving anthracyclines, for early detection and intervention. In resource-limited regions such as NWO, these insights carry significant clinical implications. Tailoring imaging to patient-specific risk factors may reduce patient burden and alleviate strain on the healthcare system by mitigating the consequences of advanced cardiovascular disease. Future research should stratify the yield based on proactive versus reactive changes in care, explore patient outcomes following changes in care, develop proactive imaging guidelines, and gather more robust data on trastuzumab-only patients. By optimizing cardiac imaging protocols in breast cancer care, healthcare resources can be allocated more effectively, ultimately improving outcomes in rural and underserved communities.

## Figures and Tables

**Figure 1 cancers-17-01909-f001:**
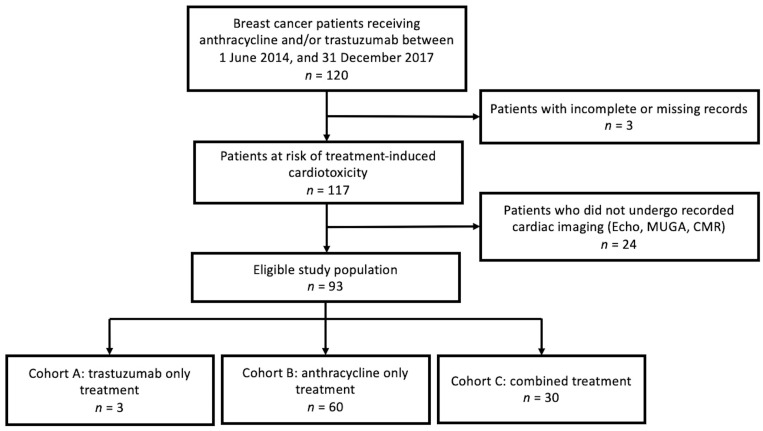
Study flow diagram. Echocardiography (Echo), multi-gated acquisition (MUGA), cardiac magnetic resonance imaging (CMR).

**Figure 2 cancers-17-01909-f002:**
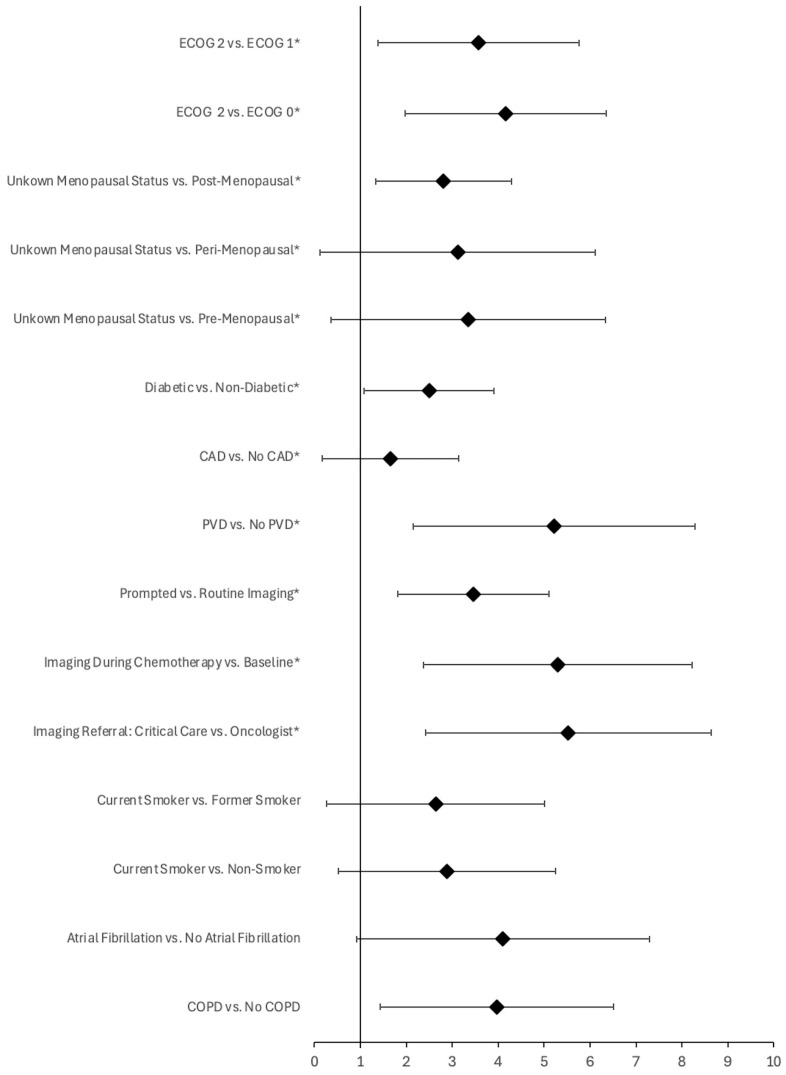
Forest plot of pre-treatment and imaging variables significantly associated with a clinically actionable change in care following cardiac imaging for patients receiving anthracyclines (cohort B, denoted with an asterisk) or combined therapy (cohort C). Error bars represent 95% confidence interval of the odds ratio (denoted with a diamond). Variables are statistically significant at *p* < 0.05.

**Table 1 cancers-17-01909-t001:** Demographic information and disease characteristics of each treatment cohort.

Characteristic	Total (*n* = 93)	Cohort A (*n* = 3)	Cohort B (*n* = 60)	Cohort C (*n* = 30)
Age (years)				
Mean (SD)	59.48 (10.41)	73.55 (9.90)	58.83 (9.83)	59.37 (10.91)
Range	35.74–84.97	67.32–84.97	35.74–81.57	39.30–75.43
Menopausal status				
Premenopausal, *n* (%)	13 (13.98)	0 (0.00)	8 (13.33)	5 (16.67)
Perimenopausal, *n* (%)	8 (8.60)	0 (0.00)	6 (10.00)	2 (6.67)
Postmenopausal, *n* (%)	65 (69.89)	2 (66.67)	41 (68.33)	22 (73.33)
Unknown, *n* (%)	7 (7.53)	1 (33.33)	5 (8.33)	1 (3.33)
BMI (kg/m^2^)				
Mean (SD)	30.38 (7.07)	24.61 (1.90)	30.31 (7.59)	31.10 (6.10)
Cancer type				
Invasive ductal carcinoma, *n* (%)	91 (97.85)	3 (100.00)	59 (98.33)	29 (96.67)
Lobular carcinoma, *n* (%)	1 (1.08)	0 (0.00)	1 (1.67)	0 (0.00)
Ductal carcinoma in situ, *n* (%)	1 (1.08)	0 (0.00)	0 (0.00)	1 (3.33)
Cancer side				
Right, *n* (%)	49 (52.69)	1 (33.33)	32 (53.33)	16 (53.33)
Left, *n* (%)	43 (46.24)	2 (66.67)	28 (46.67)	13 (43.33)
Bilateral, *n* (%)	1 (1.08)	0 (0.00)	0 (0.00)	1 (3.33)
Tumour stage				
Stage I, *n* (%)	23 (24.73)	2 (66.67)	15 (25.00)	6 (20.00)
Stage II, *n* (%)	48 (51.61)	1 (33.33)	28 (46.67)	19 (63.33)
Stage III, *n* (%)	13 (13.98)	0 (0.00)	12 (20.00)	1 (3.33)
Unknown, *n* (%)	9 (9.68)	0 (0.00)	5 (8.33)	4 (13.33)
Receptor status				
ER+, *n* (%)	65 (69.89)	2 (66.67)	40 (66.67)	23 (76.67)
PR+, *n* (%)	58 (62.37)	2 (66.67)	38 (63.33)	18 (60.00)
HER2+, *n* (%)	31 (34.41)	3 (100.00)	0 (0.00)	29 (96.67)
BRCA status				
BRCA1+, *n* (%)	2 (2.15)	0 (0.00)	2 (3.33)	0 (0.00)
BRCA2+, *n* (%)	2 (2.15)	1 (33.33)	1 (1.67)	0 (0.00)
BRCA1/2−, *n* (%)	15 (16.13)	0 (0.00)	13 (21.67)	2 (6.67)
Unknown, *n* (%)	74 (79.57)	2 (66.67)	44 (73.33)	28 (93.33)
Treatment modality				
Surgery ^a^, *n* (%)	91 (97.85)	3 (100.00)	58 (96.67)	30 (100.00)
Anthracycline, *n* (%)	90 (96.77)	0 (0.00)	60 (100.00)	30 (100.00)
Trastuzumab, *n* (%)	33 (35.48)	3 (100.00)	0 (0.00)	30 (100.00)
Endocrine therapy ^b^, *n* (%)	65 (69.89)	2 (66.67)	41 (68.33)	22 (73.33)
Radiation, *n* (%)	78 (83.87)	2 (66.67)	52 (86.67)	24 (80.00)

^a^ Surgery includes lumpectomy and mastectomy. ^b^ Endocrine therapy includes oophorectomy, GnRH agonists, tamoxifen, and aromatase inhibitors.

**Table 2 cancers-17-01909-t002:** Primary outcomes regarding the yield of cardiac imaging for breast cancer patients undergoing treatment with trastuzumab and/or anthracyclines.

Comparison	Cohort	Total Scans ^a^	Total Changes ^b^	Yield of Cardiac Imaging (95% CI)	Scans per Patient (95% CI)	Yield of Cardiac Imaging (*p*-Value ^c,d^)	Scans per Patient (*p*-Value ^c,d^)
-	Cohort A	14	1	7.14% (1.27, 31.47)	4.67 (−0.50, 9.84)	-	-
Cohort B vs. Cohort C	Cohort B	75	10	13.33% (7.41, 22.83)	1.25 (1.10, 1.40)	0.013	<0.001
	Cohort C	144	6	4.17% (1.92, 8.79)	4.80 (4.18, 5.42)	-	-

^a^ Includes recorded MUGA and Echo, no recorded instances of CMR. ^b^ Includes changes in chemotherapy regimen, change in care provider, and change in medication. Note: each scan could result in multiple changes in care. ^c^ *p*-values were calculated using chi-square exact tests for categorical variables. ^d^ Subsequent pairwise comparisons were calculated using Z-tests with the Bonferroni correction.

**Table 3 cancers-17-01909-t003:** Secondary outcomes regarding the significance of continuous data across three treatment cohorts.

Variable	Cohort	Change, Mean ± SD (Range)	No Change, Mean ± SD (Range)	*p*-Value	T-Statistic
Age (years)	A	84.97 ^a^	67.84 ± 0.73 (67.32–68.36)	-	-
	B	62.08 ± 10.05 (47.93–74.32)	58.54 ± 9.85 (35.74–81.57)	0.49	0.76
	C	62.40 ± 6.67 (52.70–71.29)	58.76 ± 11.58 (39.30–75.43)	0.36	0.96
BMI (kg/m^2^)	A	26.64 ^a^	23.60 ± 1.05 (22.86–24.34)	-	-
	B	36.89 ± 16.69 (19.88–57.45)	29.71 ± 6.16 (20.52–54.30)	0.54	- ^c^
	C	31.36 ± 3.70 (26.11–36.57)	31.04 ± 6.52 (22.14–48.89)	0.88	0.15
LVEF (%) ^b^	A	51 ^a^	65 ^a^	-	-
	B	43.67 ± 13.32 (29–55)	64.96 ± 6.96 (52–80)	0.02	−3.53
	C	44.75 ± 4.79 (38–49)	62.95 ± 7.72 (44–75)	<0.001	−6.90

^a^ *n* = 1 for this group, calculation of mean, standard deviation, range, and statistical significance not applicable. ^b^ This is LVEF associated with the scan that resulted in a change in care. ^c^ T-test not performed as data was not normally distributed per Shapiro–Wilk testing, a Mann–Whitney test was preformed instead, and U = 161.00.

## Data Availability

The raw data supporting the conclusions of this article will be made available by the authors on request.

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
