# Peer review of "Evaluating the Necessity and Impact of Cardiac Imaging on Breast Cancer Care in Northwestern Ontario"

_cancers, 2025, doi:10.3390/cancers17121909_

Round 1
Reviewer 1 Report
Comments and Suggestions for Authors
This retrospective cohort study examines the clinical utility of cardiac imaging in breast cancer patients undergoing chemotherapy with anthracyclines, trastuzumab, or both. Conducted at a single regional cancer centre in Northwestern Ontario, the analysis involves 93 patients treated between 2012 and 2017. The key objective is to assess the "yield" of cardiac imaging—defined as changes in clinical care resulting from imaging findings—across three treatment cohorts. The Authors report that the imaging yield was highest in the anthracycline-only cohort (13.3%), despite the fewest scans per patient. Trastuzumab-containing cohorts, which had more frequent imaging, exhibited lower yields (4.17%–7.14%). Predictive factors for higher imaging yield varied across cohorts but generally included lower LVEF, comorbidities (e.g., diabetes, CAD, COPD), and imaging prompted by clinical concerns rather than routine practice.The study suggests current imaging protocols may be inefficient and overly reliant on treatment regimens rather than individualized risk profiles. The manuscript presents a relevant and timely inquiry into optimizing cardiac imaging strategies in oncology, with clear implications for clinical practice, especially in resource-limited regions. The statistical methodology is generally sound, and the discussion is well-aligned with the results. However, there are several limitations and areas for improvement in structure, clarity, and analytical depth that should be addressed prior to publication. The paper is suitable for eventual publication in a journal like Cancers, but only after a thorough revision.
First, the authors must address the small and uneven cohort sizes, particularly the trastuzumab-only group (n = 3), which is statistically underpowered and excluded from most secondary analyses. While the authors acknowledge this limitation, it warrants a more rigorous treatment: consider excluding this cohort altogether from certain summary statistics or re-framing it as a pilot exploration with qualitative commentary only. The conclusions drawn from this group as a comparator are currently overstated.
Second, the definition of “yield” as “number of changes arising from scans” is potentially problematic. This metric conflates process with outcome; a more robust approach might distinguish between imaging that led to clinically actionable findings versus those that merely prompted protocol-driven changes without demonstrable patient benefit. A clearer operational definition of “change in clinical care” is needed—particularly differentiating between proactive medication initiation and reactive discontinuation of therapy.
Third, there is limited granularity in the analysis of imaging modalities. While echocardiography and MUGA are both mentioned, their relative diagnostic contributions are not disaggregated. If data permits, stratifying yield by imaging modality would enhance clinical relevance, particularly given the growing use of cardiac MRI and strain imaging.
The discussion section, while well-referenced and coherent, tends toward speculative extrapolation. For instance, the hypothesis that cumulative toxicity in the combined therapy cohort might obscure the impact of individual risk factors like diabetes is plausible but unsupported by multivariable analysis. Without formal modeling (e.g., logistic regression), such interpretations remain tentative. The authors might either conduct such analyses or explicitly state the need for future prospective studies to confirm these patterns.
Further, while the statistical methodology is generally appropriate, the manuscript would benefit from greater transparency around handling of missing data, particularly in variables like menopausal status and BRCA status. Were imputation techniques used, or were incomplete records excluded from specific analyses?
The visual presentation of data (e.g., Figure 2) is promising but underdeveloped. The authors should consider including forest plots or multivariable models summarizing effect sizes and confidence intervals for key predictors of imaging yield. This would improve both clarity and impact.
From a writing and editorial perspective, the manuscript would benefit from additional polish:
- Redundancies in the abstract and introduction should be trimmed.
- Some terminology, such as “yield,” should be defined upfront with precision;
- Paragraph transitions, particularly in the discussion, should be smoothed for logical flow.
- Minor errors (e.g., inconsistent cohort labeling, typographical issues) need correction.
Lastly, while the focus on a rural Canadian setting is commendable and adds contextual relevance, the generalizability of the findings should be more critically discussed. Are the imaging practices described consistent with national or international standards? How might these findings translate to larger, urban cancer centres or other countries?
Reviewer 2 Report
Comments and Suggestions for Authors
This is a well-designed and clinically relevant retrospective cohort study evaluating the clinical utility of cardiac imaging in breast cancer patients undergoing anthracycline and/or trastuzumab therapy.
However, a few minor issues should be addressed before publication:
-
Some lengthy and complex sentences could be simplified for clarity.
-
Certain terminology choices (e.g., “mismatch”) would benefit from replacement with more precise academic expressions.
- Figure and table citation formatting should be standardized to lowercase style throughout the Discussion.
-
While t-tests were used for continuous variable comparisons, the authors did not indicate whether normality testing (e.g., Shapiro-Wilk test) was performed to justify the use of parametric tests, especially given the small and unequal sample sizes. This should be addressed or clarified in the Statistical Analysis section.
- Although cohort A (trastuzumab-only, n=3) was excluded from secondary analyses, its extremely small sample size may still introduce bias when comparing inter-cohort results for the primary outcome. It is recommended that the authors explicitly acknowledge this limitation in the Discussion.
Round 2
Reviewer 1 Report
Comments and Suggestions for Authors
The authors have addressed all the points raised during the previous review round in a satisfactory and comprehensive manner. The manuscript has been notably improved in terms of clarity, methodological rigor, and overall structure. I believe it is now suitable for publication in its current form.
Reviewer 2 Report
Comments and Suggestions for Authors
No